# Mental health and risk of death and hospitalization in COVID–19 patients. Results from a large-scale population-based study in Spain

Aida Moreno-Juste[1,2,3☉], Beatriz Poblador-Plou[1,3☉], Cristina Ortega-Larrodé[4], Clara Laguna-Berna[1], Francisca González-Rubio[1], Mercedes Aza-Pascual-Salcedo[1,3,5], Kevin Bliek-Bueno[1,3,6], María Padilla[3,7], Concepción de-la-Cámara[4,8,9], Alexandra Prados-Torres[1,3], Luis A. Gimeno-Feliú[1,2,3,9‡], Antonio Gimeno-Miguel[1,3‡]*

1 EpiChron Research Group, Aragon Health Sciences Institute (IACS), IIS Aragon, Miguel Servet University Hospital, Zaragoza, Spain, 2 San Pablo Primary Care Health Centre, Aragon Health Service (SALUD), Zaragoza, Spain, 3 Network for Research on Chronicity, Primary Care, and Health Promotion (RICAPPS), Institute of Health Carlos III (ISCIII), Madrid, Spain, 4 Department of Psychiatry, Hospital Clínico Universitario Lozano Blesa, Zaragoza, Spain, 5 Primary Care Pharmacy Service Zaragoza III, Aragon Health Service (SALUD), Zaragoza, Spain, 6 Department of Preventive Medicine, Hospital Universitario Príncipe de Asturias, Alcalá de Henares, Madrid, Spain, 7 Research Unit, Costa del Sol Hospital, Instituto de Investigación Biomédica de Málaga (IBIMA), Marbella, Spain, 8 Centro de Investigación Biomédica en Red de Salud Mental (CIBERSAM), Ministry of Science and Innovation, Madrid, Spain, 9 Department of Medicine, Dermatology and Psychiatry, University of Zaragoza, Zaragoza, Spain

☉ These authors have contributed equally to this work as first co-authors.
‡ LAGF and AGM also contributed equally to this work as senior co-authors.
* agimenomi.iacs@aragon.es

**Data Availability Statement:** The data used in this study cannot be publicly shared because of

## Abstract

The COVID–19 pandemic has created unprecedented challenges for health care systems globally. This study aimed to explore the presence of mental illness in a Spanish cohort of COVID-19-infected population and to evaluate the association between the presence of specific mental health conditions and the risk of death and hospitalization. This is a retrospective cohort study including all individuals with confirmed infection by SARS-CoV-2 from the PRECOVID (Prediction in COVID–19) Study (Aragon, Spain). Mental health illness was defined as the presence of schizophrenia and other psychotic disorders, anxiety, cognitive disorders, depression and mood disorders, substance abuse, and personality and eating disorders. Multivariable logistic regression models were used to examine the likelihood of 30-day all-cause mortality and COVID–19 related hospitalization based on baseline demographic and clinical variables, including the presence of specific mental conditions, by gender. We included 144,957 individuals with confirmed COVID–19 from the PRECOVID Study (Aragon, Spain). The most frequent diagnosis in this cohort was anxiety. However, some differences were observed by sex: substance abuse, personality disorders and schizophrenia were more frequently diagnosed in men, while eating disorders, depression and mood, anxiety and cognitive disorders were more common among women. The presence of mental illness, specifically schizophrenia spectrum and cognitive disorders in men, and depression and mood disorders, substance abuse, anxiety and cognitive and personality disorders in

restrictions imposed by the data owner (i.e., Aragon Health Sciences Institute -IACS) due to the existence of potentially identifying patient information. This restriction has been asserted by the Clinical Research Ethics Committee of Aragon (CEICA). The authors who accessed the data belong to the EpiChron Research Group of IACS, and received permission from IACS to utilize the data for this specific study, thus implying its exclusive use by the researchers appearing in the project protocol approved by CEICA. The EpiChron Group can establish future collaborations with other groups based on the same data. However, each new project based on these data must be previously submitted to the CEICA to obtain the respective mandatory approval. Potential collaborations should be addressed to the Principal Investigator of the EpiChron Research Group, Antonio Gimeno Miguel at agimenomi. iacs@aragon.es. Requests for the data set used in this study should be addressed to CEICA at ceica@aragon.es.

**Funding:** This research was funded by the Carlos III Institute of Health, Ministry of Science and Innovation (Spain), through the Network for Research on Chronicity, Primary Care, and Health Promotion (RICAPPS) awarded on the call for the creation of Health Outcomes-Oriented Cooperative Research Networks (grant number RD21/0016/ 0019), and by Gobierno de Aragón (grant number B01_23R) and co-funded with European Union's NextGenerationEU funds. The funders had no role in study design, data collection and analysis, decision to publish, or preparation of the manuscript.

**Competing interests:** C. De-la-Camara received financial support to attend scientific meetings from Janssen, Almirall, Lilly, Lundbeck, Rovi, Esteve, Novartis, Astrazeneca, Pfizer and Casen Recordati. This does not alter our adherence to PLOS ONE policies on sharing data and materials. However, the data used in this study cannot be publicly shared because of restrictions imposed by the Aragon Health Sciences Institute and the Clinical Research Ethics Committee of Aragon. The rest of the authors declare no conflict of interest.

women, increased the risk of mortality or hospitalization after COVID–19, in addition to other well-known risk factors such as age, morbidity and treatment burden. Identifying vulnerable patient profiles at risk of serious outcomes after COVID–19 based on their mental health status will be crucial to improve their access to the healthcare system and the establishment of public health prevention measures for future outbreaks.

## Introduction

The World Health Organization (WHO) declared the coronavirus disease (COVID–19) pandemic as a Public Health Emergency of International Concern in March 2020 and, since then, SARS-CoV-2 has affected more than 600 million people and has caused over 6.4 million deaths worldwide [1]. The COVID–19 pandemic has created unprecedented challenges to health care systems globally. Identification of risk factors associated with poor outcomes rapidly emerged as one of the first steps necessary to optimize medical resources [2], guide clinical decision-making [2, 3], and target enhanced protective measures [3].

Risk factors of COVID–19 disease severity identified to date include male gender [3], older age, cardiovascular disease, and metabolic diseases like diabetes and obesity [3–5]. Also, socio-economic status [3, 6, 7], education level [6] and race have been associated as risk factors for COVID–19 disease severity in terms of mortality and need of hospitalization, highlighting the potential for the pandemic to deepen existing health inequalities [3].

Patients with mental health conditions are at particular risk of poor COVID–19 outcomes, both in terms of mortality [6–9] and hospital admissions [8, 10–12]. The evidence of the higher risk of negative outcomes of COVID-19 infection is scare and mental health disorders may increase the risk of infection due to cognitive impairment as a result of the condition, a reduced awareness of risks, and fewer patients being accepted into psychiatric wards [6]. Another reasons behind this risk are that patients with mental health conditions are more likely to develop diabetes, hypertension, chronic obstructive respiratory disease, end-stage kidney disease [7], altered immune function [13–15], higher rates of obesity and smoking, elevated stress, exposure to antipsychotics and anxiolytics, and factors related to help-seeking behaviour [15]. These patients are also associated with reduced access to care at the appropriate time [8] and socioeconomic deprivation, both of which are key factors in poor COVID–19 outcomes [7, 16].

The impact of mental health on COVID–19 outcomes (and that of the pandemic on mental health) has been documented in a number of studies [2, 3, 6–10, 12–14, 17, 18]. However, in most cases results are not reported by gender, despite the type and prevalence of mental health conditions usually differs greatly between men and women [19]. More large-scale studies addressing the wide spectrum of mental health conditions in different settings are still necessary to help determine the effect of SARS-CoV-2 infection, and of other potential future outbreaks related to this and other viruses, in people with specific mental health conditions. This could aid health-policy makers to prioritize public health strategies in preventive care and disease management strategies to meet their health needs [7, 13]. These needs, which are not only relevant because of the complications of the infection itself, but also because of the added limitations that these patients with mental health conditions may face in the access to health care, or due to homelessness, living in settings where the risk for contagion is high [20], lack of communication skills, cognitive impairment [8, 21] or the stigmatization of their mental health conditions [12].

In the literature, the study of the impact of the covid-19 pandemic on mortality and hospital admissions has been little studied jointly in the most prevalent mental health disorders in the population. As it has been mentioned, patients with mental health disorders may be at higher risk of suffering a COVID-19 infection with negative outcomes, influenced by clinical and sociodemographic variables, with differences by sex. This is why this study aimed to analyse the presence of mental illness in a Spanish cohort of COVID-19-infected population and to evaluate the association between the presence of specific mental health conditions and the risk of death and hospitalization in women and men.

## Materials and methods

### Design and study population

This is a retrospective cohort study that included virtually all individuals with laboratory-confirmed SARS-CoV-2 infection in the region of Aragon, Spain, with a reference population of 1.3 million inhabitants. Approximately 95% of the population are provided by the Spanish health system free-of-charge healthcare with universal coverage [22]. The data used in this study was obtained through the PRECOVID (Prediction in COVID–19) Study Cohort. That cohort was created through the linkage of real-world data from clinical-administrative databases and patients' electronic health records (EHRs). Its objective was to serve as the basis for observational studies on the epidemiology and trajectory of the COVID–19 disease [4]. Users of the public health system of all ages with a laboratory-confirmed infection by SARS-CoV-2 were consecutively included in this cohort along the pandemic, and their socio-demographic and clinical information linked at the patient level and then pseudonymized. Patients did not have the option of opting out of providing their data for use in this study. The authors who accessed the pseudonymized data belong to the EpiChron Research Group of Aragon Health Sciences Institute (IACS), and received permission from IACS to utilize the data for this specific study, thus implying its exclusive use by the researchers appearing in the project protocol approved by the Clinical Research Ethics Committee of Aragon (CEICA).

For this study, we accessed data from the PRECOVID Study Cohort on March 24, 2022 and we included all the individuals from the PRECOVID Study Cohort with confirmed SARS-CoV-2 infection between 4 March 2020 and 22 July 2021 (enrollment period). We followed patients for a minimum of 30 days from the date of inclusion in the cohort to study whether death occurred within 30 days of SARS-CoV-2 infection confirmation. We also assessed hospital admissions during the 15 days before and after infection. Patients were followed-up at most until 22 August 2021 (date of last data available in the cohort). We have analyzed hospitalization and mortality risk in patients with mental health disorders and COVID-19, because these two outcomes define infection severity following the definition of the COVID-19 Treatment Guidelines provided by National Institutes of Health [23].

In Aragon, Spain, different measures of geographic confinement and social restrictions were taken throughout this study period depending on the incidence of cases. In the first three months of the pandemic (March—May 2020) testing capacity in our region was limited at the time and the population was in lockdown. One this period finished, diagnostic tests were performed on all patients who were admitted to hospital and on symptomatic patients, and isolation of infected patients was carried out for 14 days. Contacts of positive patients were also actively studied. The use of masks was mandatory in the whole studied period.

The ethics committee endorsed the PRECOVID Study research protocol (PI20/226), and due to the use of anonymized data and the epidemiological nature of the project, it was not necessary to obtain the informed consent of the patients.

## Variables of study and data sources

For each subject, sociodemographic variables were analysed: age, sex, ethnicity (native vs. migrant), nationality, residence area (urban—those that concentrate in one of its municipalities at least 80% of the population of the area, and rural—the rest), and deprivation index of the area. This index was specifically developed for Aragon in a previous work [24], calculated at an aggregated level by basic healthcare area according to 26 socio-economic indicators including information on education, housing and neighbourhood conditions, unemployment rates, types of employment, ageing of the population, and immigration, and divided into four quartiles from least (Q1) to most (Q4) deprived. As clinical variables, we analysed all chronic disease diagnoses present at the date of infection and registered in primary care EHRs, the presence of multimorbidity (defined as the coexistence of two or more chronic conditions [15]), all drugs dispensed in community and hospital pharmacies during the month prior to infection, and the presence of polypharmacy defined as the simultaneous prescription of five or more drugs.

The registered diagnoses of chronic diseases were coded using the International Classification of Primary Care, First Edition (ICPC–1). A mapping system developed to codify temporary disability was used to mapped the codes to the International Classification of Diseases, Ninth Edition, Clinical Modification (ICD–9–CM) [25]. Then, theses diagnoses were subsequently sorted into 226 clinically relevant categories thought the utilization of the Clinical Classifications Software (CCS) for ICD–9–CM [26]. In some cases, diagnoses were renamed to facilitate their clinical interpretation (e.g., the term "mood disorders" was renamed as "depression and mood disorders"). The different diseases were classified as chronic or not chronic using the Chronic Condition Indicator software tool [27]. 153 conditions were categorized as chronic if they were coded during the last 12 months at least, and meeting one or both of the following criteria: (i)requires of ongoing interventions using medical products, services, and special equipment; (ii) entails limitations on self-care, social interactions, and independent living. All drugs were analysed at the third level (i.e., pharmacological subgroup) of their Anatomical–Therapeutic–Chemical (ATC) classification system code [28].

During the follow-up period, we analysed as outcome variables all-cause mortality within the first 30 days after SARS-CoV-2 diagnosis and the need for hospitalization (including admission to the Intensive Care Unit) during the 15 days before and after infection confirmation.

In our study, mental health disease was defined as the presence of at least one of the following conditions: schizophrenia and other psychotic disorders, anxiety, cognitive disorders, depression and mood disorders, substance abuse, and personality and eating disorders. Each of these conditions corresponded to a CCS category that was created based on the ICD–9–CM codes registered in patients' EHRs.

## Statistical analysis

We conducted a descriptive analysis of the population with COVID–19 infection according to the presence or not of mental illness; differences were assessed using t-tests and chi-square tests. Multivariable logistic regression models were used to examine the likelihood of 30-day all-cause mortality and COVID-19-related hospitalization (dependent variables) according to baseline demographic and clinical variables, including the diagnosis of specific mental conditions. A total of four models were performed, one for each outcome and gender combination. Results were presented as odds ratios (ORs) accompanied by their 95% confidence intervals (CI) and adjusted by all the variables included in the model. We used logistic regression instead of Cox regression considering we had the date of inclusion in the cohort (date of the

first confirmatory test result), but not the exact date of infection; moreover, follow-up was truncated at 30 days after the index date for all individuals.

Stata software (Version 12.0, StataCorp LLC, College Station, TX, USA) was used to conduct the analyses, and we considered statistical significance at p<0.05.

## Results

We included 144,957 patients with confirmed SARS-CoV-2 infection; of these, 28.7% were diagnosed with at least one of the mental illnesses considered in the study. Anxiety was the most frequent diagnosis (17.8% of the population; Table 1), followed by depression and mood disorders (11.0%), cognitive disorders (4.9%), substance abuse (1.4%), schizophrenia spectrum disorders (0.4%), eating disorders (0.4%), and personality disorders (0.3%). These figures were different according to gender. The majority of cases of substance abuse (77.3%), personality disorders (58.4%) and schizophrenia (53.0%) were diagnosed in men, whereas most of the cases of eating disorders (84.7%), depression and mood disorders (72.8%), anxiety disorders (68.7%) and cognitive disorders (64.5%) were women. In general, patients without mental illness were younger than those with mental illness, especially compared with patients with cognitive disorders who belonged to the oldest population.

The prevalence of migration was 14.20% of the patients with mental disorder and 20.03% of the total of patients without mental illness. The migrant´s nationality of patients with mental health disorders most frequent was Latin America (7.82% of the total of patients with mental illness), followed by 3.24% of Eastern Europe, North Africa (1.59%), European Union and North America (0.73%), Sub-Saharan Africa (0.64%), and Asian (0.17%). The most prevalent disorders in migrants was anxiety disorders (15.79%), depression (7.09%), cognitive disorders (1.55%), substance abuse (1.39%), eating disorders (0.34%) and personality disorders (0.34%).

Patients from all seven mental health groups showed a higher number of chronic diseases, and greater prevalence rates of multimorbidity and polypharmacy compared with patients without mental illness, especially those with cognitive disorders. As for multimorbidity prevalence and number of chronic diseases, patients with cognitive disorders were followed by those with personality disorders, and depression and mood disorders. The second mental health condition with the highest polypharmacy prevalence was depression and mood disorders, followed by schizophrenia spectrum disorders.

We observed that 1,386 (1.3%) patients without mental illness died and 7,018 (6.8%) were hospitalized. Out of all the patients with mental illness and COVID-19, 1,957 (4.7%) died and 5,745 (13.8%) were hospitalized during the follow-up period. The highest prevalence rates of death and hospitalization were observed in the group of cognitive disorders (14.6% and 26.2%, respectively). In addition, 10.7% of patients with schizophrenia spectrum disorders and 5.7% of those with depression and mood disorders died in the follow-up period.

The models developed to assess the risk of mortality revealed that the presence of schizophrenia spectrum disorders was the most influencing factor in men, being the risk of death 94% higher than in patients without mental illness (Table 2). Patients with cognitive disorders had 37.6% more risk of death than patients without mental illness. The likelihood of mortality increased by 11% and 7% with each year of age and additional chronic disease, respectively. On the other hand, living in the most deprived areas decreased mortality risk (OR 0.85, 95% CI 0.74–0.99).

In women (Table 2), the most influencing factor of mortality was substance abuse (OR 2.77, 95% CI 1.44–5.35). The presence of personality disorders, cognitive disorders, and depression and mood disorders was associated with a 2.05, 1.48 and 1.25 times higher risk of death, respectively. In addition, mortality risk increased with age (OR 1.12, 95% CI 1.11–1.13), each

**Table 1. Demographic and clinical characteristics of individuals with confirmed COVID-19 in Aragon, Spain.**

| Characteristics | | No mental illness [a] | Schizophrenia spectrum disorders | Anxiety disorders | Cognitive disorders | Depression and mood disorders | Substance abuse | Personality disorders | Eating disorders | p-value [b] |
|---|---|---|---|---|---|---|---|---|---|---|
| Total number of patients, n (%) | | 103,365 (71.31) | 608 (0.42) | 26,047 (17.97) | 7,079 (4.88) | 15,881 (10.96) | 2,046 (1.41) | 389 (0.27) | 556 (0.38) | |
| Gender, n (col%[g]) | Women | 48,199 (46.63) | 286 (47.04) | 17,894 (68.70) | 4,567 (64.51) | 11,567 (72.84) | 464 (22.68) | 162 (41.65) | 471 (84.71) | <0.001 |
| | Men | 55,166 (53.37) | 332 (52.96) | 8,153 (31.30) | 2,512 (35.49) | 4,314 (27.16) | 1,582 (77.32) | 227 (58.35) | 85 (15.29) | |
| Age, years (mean, sd [c]) | | 37.9 (22.73) | 62.09 (20.25) | 51.47 (19.65) | 72.66 (23.37) | 61.15 (19.37) | 47.99 (19.28) | 48.84 (23.48) | 38.03 (18.57) | <0.001 |
| Ethnicity, n (col% [g]) | Native | 82,661 (79.97) | 551 (90.63) | 21,844 (83.86) | 6,667 (94.18) | 13,995 (88.12) | 1,677 (81.96) | 327 (84.06) | 466 (83.81) | <0.001 |
| | Migrant | 20,704 (20.03) | 57 (9.37) | 4,203 (16.14) | 412 (5.82) | 1,886 (11.88) | 369 (18.04) | 62 (15.94) | 90 (16.19) | |
| Nationality, n (col % [g]) | Spain | 82,661 (79.97) | 551 (90.63) | 21,844 (83.86) | 6,667 (94.18) | 13,995 (88.12) | 1,677 (81.96) | 327 (84.06) | 466 (83.81) | <0.001 |
| | Eastern Europe | 3,746 (3.62) | 14 (2.30) | 1,024 (3.93) | 62 (0.88) | 415 (2.61) | 97 (4.74) | 12 (3.08) | 19 (3.42) | |
| | Asia | 767 (0.74) | 0 | 48 (0.18) | 9 (0.13) | 17 (0.11) | 5 (0.24) | 0 | 0 | |
| | North Africa | 3,000 (2.90) | 11 (1.81) | 485 (1.86) | 33 (0.47) | 190 (1.20) | 27 (1.32) | 10 (2.57) | 7 (1.26) | |
| | Sub-Saharan Africa | 2,014 (1.95) | 7 (1.15) | 167 (0.64) | 11 (0.16) | 83 (0.52) | 19 (0.93) | 2 (0.51) | 6 (1.08) | |
| | Latin America | 10,408 (10.07) | 23 (3.78) | 2,269 (8.71) | 263 (3.72) | 1,068 (6.73) | 207 (10.12) | 36 (9.25) | 54 (9.71) | |
| | European Union and North America | 769 (0.74) | 2 (0.33) | 210 (0.81) | 34 (0.48) | 113 (0.71) | 14 (0.68) | 2 (0.51) | 4 (0.72) | |
| Residence area, n (col% [g]) | Urban | 62,742 (60.70) | 313 (51.48) | 15,941 (61.20) | 3,984 (56.28) | 9,641 (60.71) | 1,111 (54.30) | 228 (58.61) | 348 (62.59) | 0.170 |
| | Rural | 40,623 (39.30) | 295 (48.52) | 10,106 (38.80) | 3,095 (43.72) | 6,240 (39.29) | 935 (45.70) | 161 (41.39) | 208 (37.41) | |
| Deprivation index [d], n (col% [g]) | Q1 | 27,177 (26.29) | 192 (31.58) | 6,719 (25.80) | 2,123 (29.99) | 4,287 (26.99) | 441 (21.55) | 114 (29.31) | 135 (24.28) | 0.915 |
| | Q2 | 24,425 (23.63) | 115 (18.91) | 6,199 (23.80) | 1,603 (22.64) | 3,793 (23.88) | 449 (21.95) | 81 (20.82) | 114 (20.50) | |
| | Q3 | 21,150 (20.46) | 133 (21.88) | 5,202 (19.97) | 1,494 (21.10) | 3,170 (19.96) | 444 (21.70) | 70 (17.99) | 112 (20.14) | |
| | Q4 | 30,613 (29.62) | 168 (27.63) | 7,927 (30.73) | 1,859 (26.26) | 4,631 (29.16) | 712 (34.80) | 124 (31.88) | 195 (35.07) | |
| No. of chronic diseases (mean, sd [c]) | | 2.06 (2.22) | 6.06 (3.64) | 5.38 (3.21) | 7.63 (3.65) | 6.59 (3.48) | 5.05 (3.11) | 5.68 (3.32) | 5.36 (3.17) | <0.001 |
| Multimorbidity [e], n (%) | | 50,319 (48.68) | 572 (94.08) | 24,323 (93.38) | 6,897 (97.43) | 15,325 (96.50) | 1,887 (92.23) | 376 (96.66) | 526 (94.60) | <0.001 |

(*Continued*)

**Table 1.** (Continued)

| Characteristics | | No mental illness [a] | Schizophrenia spectrum disorders | Anxiety disorders | Cognitive disorders | Depression and mood disorders | Substance abuse | Personality disorders | Eating disorders | p-value [b] |
|---|---|---|---|---|---|---|---|---|---|---|
| Polypharmacy [f], n (%) | | 7,928 (7.67) | 181 (29.77) | 6,069 (23.30) | 2,775 (39.20) | 5,316 (33.47) | 387 (18.91) | 92 (23.65) | 114 (20.50) | <**0.001** |

[a] Patients without studied mental diseases;

[b] p-value less than 0.05 was statistically significant, and it corresponds to the comparison of patients with mental illness vs. patients without mental illness;

[c] standard deviation;

[d] This index was specifically developed for Aragon in a previous work [23], calculated at an aggregated level by basic healthcare area according to 26 socio-economic indicators including information on education, housing and neighbourhood conditions, unemployment rates, types of employment, ageing of the population, and immigration, and divided into four quartiles from least (Q1) to most (Q4) deprived;

[e] defined as the presence of two or more chronic diseases;

[f] Defined as the dispensation of five or more drugs;

[g] The percentage is calculated per column. Statistically significant p-values are highlighted in bold.

additional chronic disease (OR 1.05, 95% CI 1.03–1.07), and the presence of polypharmacy (OR 1.04, 95% CI 1.01–1.06).

None of the mental health diagnoses studied affected the likelihood of COVID-19-related hospitalization in men (Table 3). In contrast, the risk of hospitalization increased by 27.8% in patients living in urban areas, and by 5.8% with each year of age. Multimorbidity increased the likelihood of hospitalization by 20%, and by 4.3% with each additional chronic disease (OR 1.04, 95% CI 1.03–1.06) and the presence of polypharmacy (OR 1.04, 95% CI 1.02–1.05). About the influence of nationality, we only observed that Latin America had more risk of hospitalization by 52%. Patients living in most deprived area had less risk of hospitalization and this risk increased in the less deprived areas.

Presence of schizophrenia spectrum disorders were the most influencing factor of hospitalization in women (Table 3), being the likelihood of hospital admission up to 2.4 times higher compared with women without mental illness. As in men, multimorbidity, each additional chronic disease, age and the presence of polypharmacy increased the risk of hospital admission (by 14.2%, 6.9%, 5.1% and 3.8%, respectively) in women. Living in urban areas (OR 1.30, 95% CI 1.22–1.38) also increased this risk, and patients living in Q2 areas were at higher risk than those from less deprived areas. In this group we observed that all nationalities, with the exception of Europe Union (EU) and North America, had more risk of hospitalization than Spanish population.

## Discussion

### Principals results and comparison with other studies

This population-based study in a region of Spain described the main socio-demographic and clinical characteristics of individuals with SARS-CoV-2 infection and the most prevalent mental health conditions, observing differences in the risk of death and hospitalization in men and women.

In our cohort of confirmed COVID–19 cases, mental illness was more frequent in women than in men, as it has been shown in previous similar studies [20, 29]. Anxiety was the most frequent mental condition, followed by depression and mood disorders, and cognitive disorders, although some gender-differences in the distribution of mental illnesses were shown. Wang et al. observed that patients with a recent diagnosis of a mental health condition had

**Table 2. Models of likelihood of 30-day all-cause mortality according to baseline demographic and clinical variables in COVID–19, by gender.**

**Men**

| | | OR [a] | 95% CI [b] Lower Limit | 95% CI [b] Upper Limit | p-value[c] |
|---|---|---|---|---|---|
| **Type of mental disease** | | | | | |
| No mental illness | | Reference | | | |
| Schizophrenia spectrum disorders | | 1.941 | 1.013 | 3.720 | **0.046** |
| Anxiety disorders | | 0.918 | 0.754 | 1.118 | 0.394 |
| Cognitive disorders | | 1.376 | 1.170 | 1.619 | **<0.001** |
| Depression and mood disorders | | 1.160 | 0.989 | 1.361 | 0.068 |
| Substance abuse | | 1.154 | 0.830 | 1.606 | 0.395 |
| Personality disorders | | 1.029 | 0.480 | 2.203 | 0.942 |
| Eating disorders | | - | - | - | - |
| **Age** | | **1.114** | 1.109 | 1.120 | **<0.001** |
| **Number of chronic diseases** | | **1.065** | 1.045 | 1.087 | **<0.001** |
| **Multimorbidity** | No | Reference | | | |
| | Yes | 1.212 | 0.929 | 1.582 | 0.157 |
| **Polypharmacy** | No | Reference | | | |
| | Yes | 1.024 | 1.000 | 1.049 | 0.055 |
| **Ethnicity** | Native | Reference | | | |
| | Migrant | 0.784 | 0.307 | 2.001 | 0.611 |
| **Nationality** | Spain | Reference | | | |
| | Eastern Europe | 1.157 | 0.335 | 3.991 | 0.818 |
| | Asia | 1.941 | 0.430 | 8.762 | 0.388 |
| | North Africa | 1.742 | 0.573 | 5.301 | 0.328 |
| | Sub-Saharan Africa | 0.316 | 0.036 | 2.792 | 0.300 |
| | Latin America | 0.970 | 0.330 | 2.852 | 0.956 |
| | EU[d] and North America | - | - | - | - |
| **Residence area** | Rural | Reference | | | |
| | Urban | 1.062 | 0.954 | 1.184 | 0.272 |
| **Deprivation index** | Q1 | Reference | | | |
| | Q2 | 1.019 | 0.879 | 1.182 | 0.801 |
| | Q3 | 0.949 | 0.814 | 1.107 | 0.508 |
| | Q4 | **0.854** | 0.738 | 0.988 | **0.034** |

**Women**

| | | OR [a] | 95% CI [b] Lower Limit | 95% CI [b] Upper Limit | p-value[c] |
|---|---|---|---|---|---|
| **Type of mental disease** | | | | | |
| No mental illness | | Reference | | | |
| Schizophrenia spectrum disorders | | 2.223 | 0.959 | 5.155 | 0.063 |
| Anxiety disorders | | 0.998 | 0.821 | 1.213 | 0.985 |
| Cognitive disorders | | **1.475** | 1.256 | 1.732 | **<0.001** |
| Depression and mood disorders | | **1.252** | 1.080 | 1.451 | **0.003** |
| Substance abuse | | **2.773** | 1.437 | 5.352 | **0.002** |
| Personality disorders | | **2.047** | 1.022 | 4.103 | **0.043** |
| Eating disorders | | 1.911 | 0.849 | 4.303 | 0.118 |
| **Age** | | **1.120** | 1.113 | 1.126 | **<0.001** |
| **Number of chronic diseases** | | **1.053** | 1.033 | 1.073 | **<0.001** |
| **Multimorbidity** | No | Reference | | | |
| | Yes | 0.920 | 0.634 | 1.334 | 0.660 |

*(Continued)*

**Table 2.** (Continued)

| | | | | | |
|---|---|---|---|---|---|
| **Polypharmacy** | No | Reference | | | |
| | Yes | **1.038** | 1.014 | 1.062 | **0.002** |
| **Ethnicity** | Native | Reference | | | |
| | Migrant | 0.446 | 0.133 | 1.498 | 0.192 |
| **Nationality** | Spain | Reference | | | |
| | Eastern Europe | 2.167 | 0.452 | 10.393 | 2.167 |
| | Asia | - | - | - | - |
| | North Africa | - | - | - | - |
| | Sub-Saharan Africa | 1.760 | 0.173 | 17.910 | 1.760 |
| | Latin America | 1.256 | 0.324 | 4.870 | 1.256 |
| | EU[d] and North America | - | - | - | - |
| **Residence area** | Rural | Reference | | | |
| | Urban | 1.004 | 0.899 | 1.121 | 0.940 |
| **Deprivation index** | Q1 | Reference | | | |
| | Q2 | 1.029 | 0.884 | 1.197 | 0.714 |
| | Q3 | 1.144 | 0.981 | 1.334 | 0.085 |
| | Q4 | 0.929 | 0.802 | 1.076 | 0.327 |

[a] Odds ratio adjusted by the rest of variables included in the model;

[b] Confidence Interval;

[c] p-value less than 0.05 was statistically significant. Statistically significant ORs and their p-values are highlighted in bold;

[d]EU: Europe Union.

significantly higher odds of COVID–19 infection than those free of mental illness, with depression and schizophrenia as the strongest predictors after adjusting for age, gender and ethnicity [20]. Another study showed, however, that patients with a previous diagnosis of a mental health condition had the same risk for testing positive for SARS-CoV-2 infection than people without mental illness [8].

Most of the patients in our cohort were natives; however, the higher proportion of migrants observed in the population without mental illness may be due to the "healthy migrant effect", which suggests that migrants are initially healthier than natives, and that their health worsens as the duration of their stay increases [30–34]. In addition, the lower diagnosis of mental health disorders among the migrant population is also related to immigrants´ health disadvantage explained by their experience of discrimination, access barriers to care (language culture, legal, etc.) and directly to the inherent stress of being an immigrant [30].

We found no association between deprivation index and residence area and COVID–19 infection. In contrast, in a previous study conducted in Aragon, COVID–19 incidence was slightly higher in patients belonging to the more deprived areas [29], which was associated with health areas' inequalities due to precarious conditions involving overcrowded accommodation and limited access to outdoor spaces, with more risk of exposure to the virus and different susceptibility to infection [29, 35].

In general, patients with mental illness showed a higher number of chronic diseases and multimorbidity and polypharmacy rates, which has already been related to lower incomes and educational levels [12], and a higher burden of cardiometabolic and respiratory comorbidities such as diabetes and chronic obstructive pulmonary disease (COPD) that reduce life expectancy [16].

**Table 3. Models of likelihood of COVID-19 related hospitalization according to baseline demographic and clinical variables in COVID-19, by gender.**

**Men**

| | | OR [a] | 95% CI [b] Lower Limit | 95% CI [b] Upper Limit | p-value[c] |
|---|---|---|---|---|---|
| **Type of mental disease** | | | | | |
| No mental illness | | Reference | | | |
| Schizophrenia spectrum disorders | | 1.179 | 0.775 | 1.795 | 0.441 |
| Anxiety disorders | | 0.964 | 0.878 | 1.057 | 0.435 |
| Cognitive disorders | | 0.969 | 0.857 | 1.096 | 0.618 |
| Depression and mood disorders | | 0.991 | 0.900 | 1.091 | 0.852 |
| Substance abuse | | 1.013 | 0.861 | 1.193 | 0.873 |
| Personality disorders | | 1.006 | 0.654 | 1.547 | 0.979 |
| Eating disorders | | 0.654 | 0.224 | 1.920 | 0.442 |
| **Age** | | **1.058** | 1.056 | 1.060 | **<0.001** |
| **Number of chronic diseases** | | **1.043** | 1.030 | 1.056 | **<0.001** |
| **Multimorbidity** | No | Reference | | | |
| | Yes | **1.202** | 1.103 | 1.310 | **<0.001** |
| **Polypharmacy** | No | Reference | | | |
| | Yes | **1.038** | 1.022 | 1.054 | **<0.001** |
| **Ethnicity** | Native | Reference | | | |
| | Migrant | 1.309 | 0.961 | 1.782 | 0.087 |
| **Nationality** | Spain | Reference | | | |
| | Eastern Europe | 1.378 | 0.974 | 1.950 | 0.070 |
| | Asia | 1.517 | 0.950 | 2.423 | 0.081 |
| | North Africa | 1.031 | 0.721 | 1.475 | 0.867 |
| | Sub-Saharan Africa | 1.180 | 0.810 | 1.720 | 0.388 |
| | Latin America | **1.520** | 1.100 | 2.101 | **0.011** |
| | EU[d] and North America | - | - | - | - |
| **Residence area** | Rural | Reference | | | |
| | Urban | **1.278** | 1.207 | 1.354 | **<0.001** |
| **Deprivation index** | Q1 | Reference | | | |
| | Q2 | **1.113** | 1.031 | 1.203 | **0.006** |
| | Q3 | 0.994 | 0.916 | 1.078 | 0.881 |
| | Q4 | **0.910** | 0.844 | 0.981 | **0.014** |

**Women**

| | | OR [a] | 95% CI [b] Lower Limit | 95% CI [b] Upper Limit | p-value[c] |
|---|---|---|---|---|---|
| **Type of mental disease** | | | | | |
| No mental illness | | Reference | | | |
| Schizophrenia spectrum disorders | | **2.391** | 1.505 | 3.797 | **<0.001** |
| Anxiety disorders | | 0.920 | 0.844 | 1.004 | 0.060 |
| Cognitive disorders | | 0.904 | 0.808 | 1.011 | 0.076 |
| Depression and mood disorders | | 1.011 | 0.934 | 1.095 | 0.783 |
| Substance abuse | | 1.362 | 0.964 | 1.925 | 0.080 |
| Personality disorders | | 1.229 | 0.745 | 2.026 | 0.419 |
| Eating disorders | | 0.961 | 0.633 | 1.459 | 0.851 |
| **Age** | | **1.051** | 1.049 | 1.053 | **<0.001** |
| **Nº of chronic diseases** | | **1.069** | 1.057 | 1.081 | **<0.001** |
| **Multimorbidity** | No | Reference | | | |
| | Yes | **1.143** | 1.024 | 1.277 | **0.018** |

(*Continued*)

**Table 3.** (Continued)

| | | | | | |
|---|---|---|---|---|---|
| **Polypharmacy** | No | Reference | | | |
| | Yes | **1.038** | 1.024 | 1.052 | **<0.001** |
| **Ethnicity** | Native | Reference | | | |
| | Migrant | 1.090 | 0.742 | 1.601 | 0.662 |
| **Nationality** | Spain | Reference | | | |
| | Eastern Europe | **1.630** | 1.074 | 2.474 | **0.022** |
| | Asia | **2.807** | 1.635 | 4.817 | **0.000** |
| | North Africa | **1.712** | 1.101 | 2.661 | **0.017** |
| | Sub-Saharan Africa | **2.291** | 1.438 | 3.649 | **0.000** |
| | Latin America | **1.775** | 1.197 | 2.633 | **0.004** |
| | EU[d] and North America | - | - | - | - |
| **Residence area** | Rural | Reference | | | |
| | Urban | **1.300** | 1.223 | 1.382 | **<0.001** |
| **Deprivation index** | Q1 | Reference | | | |
| | Q2 | **1.123** | 1.037 | 1.217 | **0.005** |
| | Q3 | 0.988 | 0.907 | 1.077 | 0.878 |
| | Q4 | 0.964 | 0.891 | 1.046 | 0.350 |

[a] Odds ratios adjusted by the rest of variables included in the model;

[b] Confidence Interval;

[c] p-value less than 0.05 was statistically significant. Statistically significant ORs and their p-values are highlighted in bold;

[d] EU: Europe Union.

One of the major findings of this cohort study was that mental health conditions were associated to an increase in the risk of mortality and hospitalization in COVID–19 patients, and that their effect depended on gender. Gender is one of the determinants of health that produces considerable consequences for health outcomes [20]. Mental health differences between gender have been attributed to sex and gender differences: genetic predisposition; different sex hormones; dysregulations in the hypothalamic-pituitary-adrenal axis, especially for mood disorders; low self-esteem; gender-based violence; belonging to a gender minority; and other factors like education, employment, housekeeping and socioeconomic status [36]. These differences in the prevalence of mental diseases depending on gender has been related to gender disparities in COVID-19 infection rates [20]. Through it is not an objective of our work, other studies have determined that hospitalizations and deaths are more frequent in men [20].

Although, recent studies confirmed that pre-existing mental health conditions were associated with COVID–19 mortality and an increased risk of hospitalisation [7, 14], this is one of the few studies that analyse the risk of mortality and hospitalizations considering the most frequent patterns of mental illness in the population at the same time, and differentiating by sex. The risk of COVID-19 mortality and hospitalisation has been related to patients´ high morbidity burden [20], and that it was also more frequent in patients with psychotic disorders like schizophrenia and schizotypal disorders, compared with patients with mood disorders [13, 14, 37]. The fact that cognitive disorders were the mental illness with the highest prevalence of mortality and hospitalization has been observed in other studies [2, 38]. It could be explained by the fact that patients with dementia may have less capacity to communicate their medical concerns [38], the atypical symptoms that may impede the early recognition of the disease, and their association with age and comorbid conditions [2]. Schizophrenia spectrum disorders have been associated with an increased risk of death [3, 6, 9, 10, 12, 21] and hospitalization,

and this risk was not only limited to psychiatric diagnoses in hospitals [12, 16]. In another study, schizophrenia also increased mortality; however, it did not associate an increase in admissions to the Intensive Care Unit compared with non- schizophrenia patients [10, 21]. Another one showed that schizophrenia spectrum disorders were related to mortality, while mood and anxiety disorders were not [3].

In addition, the fact that mental illness increased the risk of negative outcomes of COVID–19 infection has also been linked to the relationship between severe mental illness (SMI), unhealthy lifestyles (e.g., physical inactivity, poor diets, sleep disturbances [12, 37], reduction in self-care, social isolation, lack of a caregiver or family support [8] and high alcohol and tobacco use) and a higher burden of somatic disease [12, 37]. Furthermore, SMI has been associated with dysregulation of biological processes such as immune-inflammatory alterations, which may predispose these patients to more severe infections or to secondary bacterial infections [12, 37]. Even before COVID–19, the incidence of pneumonia was higher in SMI, and was associated with antipsychotic medications, tobacco use, and other factors. Furthermore, clozapine, which is the antipsychotic reserved for treatment-resistant schizophrenia patients, can suppress immune functions, and increase susceptibility to infections like pneumonia [38]. Toubasi and Lee et al. observed that patients who were diagnosed with mental health conditions needed more mechanical ventilation, were more frequently admitted to the Intensive Care Unit, and had higher mortality [8, 14]. Fond et al. described an increased risk of mortality and poor COVID–19 outcomes in patients with mental health conditions irrespective of the main clinical risk factors for severe COVID–19 [7]. This suggests that other factors can lead to these health inequities, like a lack of communication skills, cognitive impairment [8, 21], limited comprehension of medical advice, poor self-awareness [21], lack of caregivers or family support [8] and the stigmatization of mental health conditions, which might aggravate social isolation of patients with mental health conditions and complicate the diagnosis of COVID–19 [12]. Thus, it is possible that patients with severe SMI, may have a harder time complying with protective hygiene measures, stay-at-home advice, and other health guidance during this pandemic [10, 39], or they might reside in congregate facilities, such as psychiatric inpatient units, homeless shelters, community housings, and prisons, where risk of COVID–19 transmission is increased because of the inability to effectively socially distance and/or quarantine [13]. Long-term use of alcohol, tobacco and other drugs is associated with pulmonary (COPD, pulmonary hypertension), cardiovascular (myocardial infarction, arrhythmias and cardiac insufficiency), and metabolic (hypertension and diabetes) diseases, all of which are risk factors for COVID-19 infection and its negative outcomes [7]. In addition, tobacco, whose use is much more common in SMI [10, 20, 39], is associated with a worsening of the prognosis in COVID-19, because of its effect on the respiratory system and immune responsiveness [10, 13], and the higher angiotensin converting enzyme 2 (ACE–2) levels in the airways [39]. All these factors could result in a delay in the access to hospital care and a worsening of the respiratory condition, which are important indicators of serious illness [21].

The influence of migration as a whole variable in the risk of mortality and hospitalization was not significant, but we observed some differences analysing the different nationalities. In the case of mortality, no significant differences were observed in either gender, but in hospitalization, Latin America men had 52% more risk than Spanish, and all women from the foreign nationalities, with the exception of EU and North America, had more risk of hospitalization than Spanish women. This increase in risk of hospitalization in the migrant population has been an unexpected outcome. In general, other studies have shown that this population suffers fewer hospitalizations than the native population [40, 41]. This finding would merit specific studies aimed at finding a more detailed explanation.

About the influence of deprivation index in the risk of mortality and hospitalization, we observed differences in the models analysed. The risk of hospitalization was higher in patients living in less deprived areas. This could be explained because migrants may have more difficulties to access the health system, they do not receive adequate care in the initial stages and are treated in more serious situations requiring hospitalization [30]. On the contrary, in this study the risk of mortality decreased in most deprived areas in men. This result must be interpreted with caution since deprivation index is an ecological index calculated in an aggregated form data, so it would be necessary to examine more deeply the influence of this variable in mortality risk in specifics studies.

As we saw, in some cases, multimorbidity [9, 12, 37] and having an additional chronic disease were associated with COVID–19 complications, especially in diabetes, obesity, cardiovascular disease, COPD, immunodeficiency, cancer and hypertension [39, 42, 43], and these diseases are more common in patients with SMI [10]. It is believed that certain comorbidities increase the inflammatory response, a biological factor common to severe COVID–19 pathophysiology (cytokine storm), and the chronic low-grade inflammation caused by mental illnesses. This response could induce the development of acute respiratory distress syndrome and death in patients with COVID–19 [6, 9, 12, 20]. The reasons why underlying medical conditions cause more severe COVID–19 cases are not yet fully understood, but ACE–2, the receptor to which SARS-CoV-2 binds to cause infection, are highly expressed in the heart and lungs. Whatever the mechanism, the high rate of smoking and comorbid medical conditions in SMI, in combination with the medications routinely used to treat SMI, may create the perfect storm for COVID–19 complications [39]. In addition, polypharmacy has been found to be associated with a higher risk of developing severe COVID–19 [6]. Other studies have reported that exposure to anxiolytic and antipsychotic drug treatments was associated with higher risk of pneumonia [13] and severe COVID–19 outcomes [6, 37]. Although the cause is unclear [6], it is thought that antipsychotics precipitate cardiovascular and thromboembolic risk, interfering with an adequate immune response, and might cause pharmacokinetic and pharmacodynamic interactions with drugs used to treat COVID–19 [37].

Although further research is required to determine the underlying mechanisms of these associations, our findings highlight the need for targeted approaches to manage and prevent COVID–19 in the at-risk groups that were identified in this study [37], as well as the need for the development of new clinical decision-making guidelines, including improvements in follow-up and targeted interventions [3]. Other recommendations are to maintain active surveillance, prompt and timely treatments when needed [8, 12] and the prioritization of these risk groups during vaccination campaigns [12, 14]. Hence, an important in the prevention of unfavourable COVID–19 outcomes in these patients would be an optimal management of their pulmonary, metabolic and cardiovascular diseases, and improve their treatment adherence [12].

## Strengths and limitations

One of the main strengths of this study is the fact that we analysed virtually all the individuals of the reference population with a confirmed infection by COVID–19, and we exhaustively analysed all the chronic diagnoses registered by physicians in their EHRs. Many of the published studies focus on the study of the risk of hospitalizations or mortality, or include one or two mental illnesses. Few studies study all factors at the same time, considering the multimorbidity and sociodemographic variables. In addition, it is interesting to analysed this population-based data in a region of Spain to add comparability with the situation generated by this pandemic in other regions of the world. Furthermore, a free software with an open algorithm

was used to manage chronic conditions in the analyses, facilitating reproducibility and comparability of results. One of the main limitations lies on the observational, cross-sectional nature of the study, which prevents the establishment of causal relationships between the explanatory variables and infection severity; nonetheless, our results point to certain factors that should be further investigated to confirm their influence. Another limitation of our study was related to the lack of relevant variables such as tobacco use, body mass index, reduction in self-care, isolation from society, difficulty of communication, and social determinants (e.g lack of a caregiver or family support) that could influence the severity of COVID–19 and that were not available in our cohort. Given that smoking, alcohol use and being overweight are especially common among patients with psychiatric disorders, we cannot rule out the possibility that our results could be influenced by some of these confounding factors. Moreover, we collected data on the use of oral psychotropic drugs but the information on intravenously administered psychotropic drugs was not recorded. Finally, the COVID–19 pandemic occurred in a series of waves that, as other studies have pointed out, has led to differences in infection diagnosis due to the availability of diagnostic tests and the updates in diagnostic protocols, resulting in a potential misclassification of cases.

## Conclusions

The risk of mortality and/or hospitalization after COVID–19 infection are suggested to increase with the presence of mental illness, specifically schizophrenia spectrum and cognitive disorders in men, and depression and mood disorders, substance abuse, anxiety and cognitive and personality disorders in women, in addition to other well-known risk factors like age and morbidity and treatment burden. The identification of vulnerable patients with mental health conditions at higher risk of serious consequences of COVID–19 infection is crucial for the improvement of their access to healthcare and the establishment of public health prevention measures for future outbreaks.

## Author Contributions

**Conceptualization:** Aida Moreno-Juste, Cristina Ortega-Larrodé, Concepción de-la-Cámara, Alexandra Prados-Torres, Antonio Gimeno-Miguel.

**Data curation:** Aida Moreno-Juste, Beatriz Poblador-Plou, Clara Laguna-Berna.

**Formal analysis:** Beatriz Poblador-Plou, Clara Laguna-Berna.

**Methodology:** Beatriz Poblador-Plou.

**Project administration:** Alexandra Prados-Torres, Antonio Gimeno-Miguel.

**Supervision:** Alexandra Prados-Torres, Luis A. Gimeno-Feliú, Antonio Gimeno-Miguel.

**Validation:** Aida Moreno-Juste.

**Visualization:** Aida Moreno-Juste, Antonio Gimeno-Miguel.

**Writing – original draft:** Aida Moreno-Juste, Cristina Ortega-Larrodé.

**Writing – review & editing:** Francisca González-Rubio, Mercedes Aza-Pascual-Salcedo, Kevin Bliek-Bueno, María Padilla, Concepción de-la-Cámara, Alexandra Prados-Torres, Luis A. Gimeno-Feliú, Antonio Gimeno-Miguel.

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
