## [Decision Letter · Decision Letter 0]

2 Jan 2024

PONE-D-23-37788Mental health and risk of death and hospitalization in COVID-19 patients. Results from a large-scale population-based study in Spain.PLOS ONE

Dear Dr. Gimeno-Miguel,

Thank you for submitting your manuscript to PLOS ONE. After careful consideration, we feel that it has merit but does not fully meet PLOS ONE’s publication criteria as it currently stands. Therefore, we invite you to submit a revised version of the manuscript that addresses the points raised during the review process.

We look forward to receiving your revised manuscript.

Kind regards,

Santiago Gascón, PhD

Academic Editor

PLOS ONE

https://jamanetwork.com/journals/jamapsychiatry/fullarticle/2782453?

Poblador-Plou, B.; Carmona-Pírez, J.; Ioakeim-Skoufa, I.; Poncel-Falcó, A.; Bliek-Bueno, K.; Cano-del Pozo, M.; Gimeno-Feliú, L.A.; González-Rubio, F.; Aza-Pascual-Salcedo, M.; Bandrés-Liso, A.C.; et al. Baseline Chronic Comorbidity and Mortality in Laboratory-Confirmed COVID-19 Cases: Results from the PRECOVID Study in Spain. Int. J. Environ. Res. Public Health 2020, 17, 5171. https://doi.org/10.3390/ijerph17145171

In your revision ensure you cite all your sources (including your own works), and quote or rephrase any duplicated text outside the methods section. Further consideration is dependent on these concerns being addressed.

 [This research was funded by the Carlos III Institute of Health, Ministry of Science and Innovation (Spain), through the Network for Research on Chronicity, Primary Care, and Health Promotion (RICAPPS) awarded on the call for the creation of Health Outcomes-Oriented Cooperative Research Networks (grant number RD21/0016/0019), and by Gobierno de Aragón (grant number B01_23R) and co-funded with European Union’s NextGenerationEU funds.].  

[This research was funded by the Carlos III Institute of Health, Ministry of Science and Innovation (Spain), through the Network for Research on Chronicity, Primary Care, and Health Promotion (RICAPPS) awarded on the call for the creation of Health Outcomes-Oriented Cooperative Research Networks (grant number RD21/0016/0019), and by Gobierno de Aragón (grant number B01_23R) and co-funded with European Union’s NextGenerationEU funds.]

 [This research was funded by the Carlos III Institute of Health, Ministry of Science and Innovation (Spain), through the Network for Research on Chronicity, Primary Care, and Health Promotion (RICAPPS) awarded on the call for the creation of Health Outcomes-Oriented Cooperative Research Networks (grant number RD21/0016/0019), and by Gobierno de Aragón (grant number B01_23R) and co-funded with European Union’s NextGenerationEU funds.].

6. Thank you for stating the following in the Competing Interests section: 

[C. De-la-Camara received ﬁnancial support to attend scientiﬁc meetings from Janssen, Almirall, Lilly, Lundbeck, Rovi, Esteve, Novartis, Astrazeneca, Pfizer and Casen Recordati. The rest of the authors declare no conflict of interest.]. 

7. We note that you have indicated that there are restrictions to data sharing for this study. PLOS only allows data to be available upon request if there are legal or ethical restrictions on sharing data publicly. For more information on unacceptable data access restrictions, please see http://journals.plos.org/plosone/s/data-availability#loc-unacceptable-data-access-restrictions. 

Additional Editor Comments:

We consider that the article submitted is of interest, but the comments sent by the two reviewers should be taken into account, especially in terms of defining some concepts that are mentioned (without specifying) in different sections, and also in terms of further developing the discussion in the light of the results obtained.

Please consider each of the indications offered by the reviewers.

Reviewers' comments:

Reviewer's Responses to Questions

**Comments to the Author**

1. Is the manuscript technically sound, and do the data support the conclusions?

Reviewer #1: Partly

Reviewer #2: Yes

2. Has the statistical analysis been performed appropriately and rigorously? 

Reviewer #1: No

Reviewer #2: I Don't Know

3. Have the authors made all data underlying the findings in their manuscript fully available?

Reviewer #1: No

Reviewer #2: Yes

4. Is the manuscript presented in an intelligible fashion and written in standard English?

Reviewer #1: Yes

Reviewer #2: Yes

5. Review Comments to the Author

Reviewer #1: Peer Review

Mental health and risk of death and hospitalization in COVID-19 patients. Results from a large-scale population-based study in Spain.

Line 62: Please clarify what you mean by “risk factors”. Risk factors for what?

Line 64: You say “Also, socioeconomic status [3,6-7], education level [6] and race have been associated as risk factors for severity and mortality…”. Please indicate what is meant by severity. Symptom severity? Infection severity? Length of symptoms?

Lines 66 and 68 and throughout: The term “mental disorders” can be stigmatizing. I recommend that this be changed to “mental health conditions” or “mental health concerns” or “mental health diagnoses”.

Lines 74-75: You indicate that a number of studies have been done on the impact of mental health on COVID-19 outcomes and vice versa, but do not include citations. Please add them in.

Lines 77-84: Please revisit this text for clarity. I suggest breaking this up into multiple sentences.

Introduction: The paper feels a bit imbalanced. There are approximately 2 pages of introductory text and approximately 6 pages of discussion text. The discussion reviews previous findings, but should more clearly contextualize the findings from the current study within previous literature. The introduction should more clearly state the rationale for the study (some of this could be -pulled from the discussion) and what gap this study addresses. I note the use of the term “explore” in line 85, but I would hope that the researchers had a specific hypothesis/hypotheses that could be clearly stated, especially as they use the term “evaluate” on line 86. While the authors present interesting results, without a hypothesis/hypotheses and clearly stating the rationale/gap in the literature, it is not clear what this study adds/why it is important.

Lines 97-100: For replication purposes as well as evaluation of systematic missingness, do patients have the option of opting out of providing their data for use in studies? If so, please describe. If not, please indicate.

Lines 94, and 101-103 included acronyms that may be unfamiliar to readers. Please expand these and any others in the text upon first use (e.g., PRECOVID, IACS, CEICA, etc.).

Line 106: Typo of the word “enrollment”

Line 109: What is meant by “Follow-up ended invariably on 22 August 2021”. Why? Please explain. Was this a pre-determined date?

Lines 114-115: You collapsed ethnicity into a dichotomous (native v. migrant) variable. Please include numbers regarding the ethnic groups represented in the study (ns, percentages) in the text of the methods. As migration status may be related to the variables of interest at varying degrees per ethnic group (depending on country of origin, racial tensions, cultural traditions regarding accessing healthcare and/or number of people in household and/or how mental health conditions are viewed, health status mandates for migration, etc.), sensitivity analyses should also be done per ethnic group.

Lines 116-117, superscript d in Table 1: For replication and interpretation purposes, please cite and clearly define the deprivation index utilized in this study.

Line 127: Please articulate which diagnostic labels were renamed to facilitate clinical interpretation.

Lines 146-148: What was your reasoning for conducting all analyses by gender, as opposed to the other variables that you mentioned have previously been indicated as risk factors for poor COVID-19 outcomes (lines 62-65), or the variables with significant p-values in Table 1? If this was pre-determined (and it sounds like it was), please include this rationale in the introduction section.

Lines 146-147: The phrase, “..examine the likelihood of 30-day all-cause mortality and COVID-19 related hospitalization (dependent variable…” makes it sound like you created some sort of composite of mortality and COVID-19 hospitalization and ran one regression. Please specify that you are describing different models. You should also somewhere in this section (recommended) or elsewhere describe why it is important to look at both COVID-19 hospitalization AND all-cause mortality when drawing conclusions. Please describe how this comparison lends critical information for testing your hypotheses.

Line 150: How did you determine which covariates to include in the models?

Statistical Analysis section: How were missing data handled?

Table 1: I appreciate the inclusion of the percent representation for the migrant group within the overall category of individuals with schizophrenia. Please include additional clarification on the percent representation specifically within the migrant population. For instance, could you provide the percent of individuals with schizophrenia among migrants, based on the total number of migrants? This clarification would enhance the interpretation of the data and contribute to a more nuanced understanding of the prevalence within the migrant subgroup. I have the same comment for the other characteristics listed in Table 1.

Lines 176-177: Please also include the ns and percentages for those without mental health diagnoses and COVID-19.

Line 197: Please double check this sentence. Did you mean that “None of the mental health diagnoses studied affected the likelihood of COVID-19-related hospitalization in men.”?

Lines 228-231: I appreciate that you have described some of particulars related to migrant status as related to the “healthy migrant effect”. Please also provide alternate explanations for the migrant findings.

Discussion: For men, living in most deprived areas decreased mortality risk (14.5%) but living in a less deprived area increased hospitalization risk (11%). And being a migrant increased hospitalization risk by 74%. For women, being a migrant decreased mortality risk substantially (nearly 50%), but increased hospitalization risk by 90%. And living in a less deprived area increased hospitalization risk by 12%. These findings and potential implications need to be expounded on in the discussion.

Lines 246-260: Thank you for detailing some of the previous work on schizophrenia and COVID-19 mortality and related hospitalization. I recommend that you include a sentence or two detailing what your study adds to this literature.

Lines 261-292: From this literature, I am left questioning why you did not include variables related to alcohol and tobacco use, as well as caregiver/family support, and hygiene practices in this study. Please seriously consider including these and if it is not feasible, please describe why and how this may limit the interpretation of the results. I see that you have done so for alcohol and tobacco use and BMI (lines 329-334), but please include other limitations related to lack of inclusion of variables related to caregiver/family support, and hygiene practices.

Lines 293-296: This sentence, “As we saw, in some cases, multimorbidity….more common in patients with SMI” made me question why you did not delineate specific chronic diseases in this study. I appreciate your reliance on other literature; however, if you want to make this claim (as well as the claim in lines 318-320), I would like to see these analyses broken down by chronic disease type.

Lines 336-339: Great point regarding the COVID-19 pandemic waves. I also wonder about lockdown/quarantine procedures/recommendations in Aragon, Spain. Please include these details somewhere in the text. This will help with reader understanding of how these results may generalize (OR NOT) to other populations.

Reviewer #2: Thanks very much for the authors for choosing this important topic.

Well done overall.

However, you may flourish the discussion more around the finding related to males versus female differences . What are the possible pathophysiological, psychosocial explanations as well as implications of such finding?

6. PLOS authors have the option to publish the peer review history of their article (what does this mean?). If published, this will include your full peer review and any attached files.

Reviewer #1: No

Reviewer #2: No

---

## [Author Response · Author response to Decision Letter 0]

17 Jan 2024

Dear Editor,

Thank you very much for giving us the opportunity of improving the quality of our manuscript to be published in PLOS ONE.

Below, we answer all the journal requirements.

We have also prepared a point-by-point response letter to reviewers.

Sincerely,

Antonio Gimeno-Miguel

Thanks for your reminder. We checked the requirements and modified what is necessary to meet PLOS ONE's style requirements.

2. We noticed you have some minor occurrence of overlapping text with the following previous publication(s), which needs to be addressed: Poblador-Plou, B.; Carmona-Pírez, J.; Ioakeim-Skoufa, I.; Poncel-Falcó, A.; Bliek-Bueno, K.; Cano-del Pozo, M.; Gimeno-Feliú, L.A.; González-Rubio, F.; Aza-Pascual-Salcedo, M.; Bandrés-Liso, A.C.; et al. Baseline Chronic Comorbidity and Mortality in Laboratory-Confirmed COVID-19 Cases: Results from the PRECOVID Study in Spain. Int. J. Environ. Res. Public Health 2020, 17, 5171. https://doi.org/10.3390/ijerph17145171

Thank you. This is indeed a previous work of our research group with similar methodology. We have referenced it and made some changes to avoid overlapping text.

Thank you very much for the advice. Unfortunately, current restrictions imposed by the data owner and by the ethics committee prevent us from depositing our data in a repository as they cannot leave our government server even they are anonymized.

4. Thank you for stating the financial disclosure: Please state what role the funders took in the study.

Thank you. We have included the following statement: “The funders had no role in study design, data collection and analysis, decision to publish, or preparation of the manuscript”.

5. Thank you for stating the Acknowledgments Section. We note that you have provided funding information that is not currently declared in your Funding Statement. However, funding information should not appear in the Acknowledgments section or other areas of your manuscript. We will only publish funding information present in the Funding Statement section of the online submission form. 

Please remove any funding-related text from the manuscript and let us know how you would like to update your Funding Statement. 

Thank you. We have modified Funding Statement section as:

This research was funded by the Carlos III Institute of Health, Ministry of Science and Innovation (Spain), through the Network for Research on Chronicity, Primary Care, and Health Promotion (RICAPPS) awarded on the call for the creation of Health Outcomes-Oriented Cooperative Research Networks (grant number RD21/0016/0019), and by Gobierno de Aragón (grant number B01_23R) and co-funded with European Union’s NextGenerationEU funds. The funders had no role in study design, data collection and analysis, decision to publish, or preparation of the manuscript.

And we have removed funding information from other sections in the manuscript.

6. Thank you for stating the Competing Interests section. Please confirm that this does not alter your adherence to all PLOS ONE policies on sharing data and materials, by including the following statement: ""This does not alter our adherence to PLOS ONE policies on sharing data and materials.” (as detailed online in our guide for authors http://journals.plos.org/plosone/s/competing-interests). If there are restrictions on sharing of data and/or materials, please state these. Please note that we cannot proceed with consideration of your article until this information has been declared. 

Thank you. We have included the suggested statement in the Conflicts of interest section of the manuscript as the author´s conflict of interest did not alter our adherence to PLOS ONE policies. However, we also included, as done in Data availability section, that the data used in this study cannot be publicly shared because of restrictions imposed by the Aragon Health Sciences Institute and the Clinical Research Ethics Committee of Aragon.

7. We note that you have indicated that there are restrictions to data sharing for this study. PLOS only allows data to be available upon request if there are legal or ethical restrictions on sharing data publicly. 

As we stated in the Data availability section of the manuscript, the data used in this study cannot be publicly shared because of legal restrictions imposed by the Aragon Health Sciences Institute and the Clinical Research Ethics Committee of Aragon. We have revised the statement which should read as:

The data used in this study cannot be publicly shared because of restrictions imposed by the data owner (i.e., Aragon Health Sciences Institute -IACS) due to the existence of potentially identifying patient information. This restriction has been asserted by the Clinical Research Ethics Committee of Aragon (CEICA, ceica@aragon.es).

The authors who accessed the data belong to the EpiChron Research Group of IACS, and received permission from IACS to utilize the data for this specific study, thus implying its exclusive use by the researchers appearing in the project protocol approved by CEICA. The EpiChron Group can establish future collaborations with other groups based on the same data. However, each new project based on these data must be previously submitted to the CEICA to obtain the respective mandatory approval. Potential collaborations should be addressed to the Principal Investigator of the EpiChron Research Group, Antonio Gimeno Miguel at agimenomi.iacs@aragon.es.

8. Please review your reference list to ensure that it is complete and correct. If you have cited papers that have been retracted, please include the rationale for doing so in the manuscript text, or remove these references and replace them with relevant current references. Any changes to the reference list should be mentioned in the rebuttal letter that accompanies your revised manuscript. If you need to cite a retracted article, indicate the article’s retracted status in the References list and also include a citation and full reference for the retraction notice

We have added references 17, 18, 19 and better referenced the data appearing in lines 80-83. We also added other new references: 36, 40 and 41.

When adding new references, the numbering of the subsequent bibliographical references inserted in the manuscript has been modified. Also, we have modified the reference list named the first six authors, et al.

Reviewer #1:

Dear Reviewer,

Thank you very much for your comprehensive review of our manuscript. We believe that all the comments and suggestions received are very pertinent and useful for improving the quality and clarity of our work.

Please find below a point-by-point response to all your comments, accompanied by a revised version of the manuscript with tracked changes.

Line 62: Please clarify what you mean by “risk factors”. Risk factors for what? 

We referred to risk factors of COVID-19 disease severity in terms of risk of death and hospitalization after infection. We have clarified it in the text (Lines 83-86, revised version of the manuscript).

Line 64: You say “Also, socioeconomic status [3,6-7], education level [6] and race have been associated as risk factors for severity and mortality…”. Please indicate what is meant by severity. Symptom severity? Infection severity? Length of symptoms? 

In line with your previous comment, we referred to disease severity in terms of death and need of hospitalization. We have clarified it in the text (Lines 85-86). 

Lines 66 and 68 and throughout: The term “mental disorders” can be stigmatizing. I recommend that this be changed to “mental health conditions” or “mental health concerns” or “mental health diagnoses”. 

We totally agree with your appreciation. We have changed the term “disorders” to “health conditions” throughout the manuscript.

Lines 74-75: You indicate that a number of studies have been done on the impact of mental health on COVID-19 outcomes and vice versa, but do not include citations. Please add them in. 

We agree with your comment. We have included the references (Lines 80-81) and also added the following ones:

- G Serafini, B Parmigiani, A Amerio, A Aguglia, L Sher, M Amore, The psychological impact of COVID-19 on the mental health in the general population, QJM: An International Journal of Medicine, Volume 113, Issue 8, August 2020, Pages 531–537, https://doi.org/10.1093/qjmed/hcaa201

- Jiaqi Xiong, Orly Lipsitz, Flora Nasri, Leanna M.W. Lui, Hartej Gill, Lee Phan, David Chen-Li, Michelle Iacobucci, Roger Ho, Amna Majeed, Roger S. McIntyre, Impact of COVID-19 pandemic on mental health in the general population: A systematic review, Journal of Affective Disorders, Volume 277, 2020, https://doi.org/10.1016/j.jad.2020.08.001.

Lines 77-84: Please revisit this text for clarity. I suggest breaking this up into multiple sentences. 

We agree with your comment. We have split the text in several sentences (Lines 83-94).

Introduction: The paper feels a bit imbalanced. There are approximately 2 pages of introductory text and approximately 6 pages of discussion text. The discussion reviews previous findings, but should more clearly contextualize the findings from the current study within previous literature. The introduction should more clearly state the rationale for the study (some of this could be -pulled from the discussion) and what gap this study addresses. I note the use of the term “explore” in line 85, but I would hope that the researchers had a specific hypothesis/hypotheses that could be clearly stated, especially as they use the term “evaluate” on line 86. While the authors present interesting results, without a hypothesis/hypotheses and clearly stating the rationale/gap in the literature, it is not clear what this study adds/why it is important. 

Thank you very much for your useful comment. We have stated the gap in the literature and our hypothesis in the introduction (Lines 82-83, 95-102). We have also modified the discussion with this clarification (Lines 276-279, 320-324, 425-429).

Lines 97-100: For replication purposes as well as evaluation of systematic missingness, do patients have the option of opting out of providing their data for use in studies? If so, please describe. If not, please indicate. 

Thank you for your comment. Patients did not have the option of opting out of providing their data for use in studies. We have indicated this in the text (Line 118-119).

Lines 94, and 101-103 included acronyms that may be unfamiliar to readers. Please expand these and any others in the text upon first use (e.g., PRECOVID, IACS, CEICA, etc.). 

Thank you for the observation. Acronyms are now expanded upon first use in the text (Lines 35, 110, 121,123, 164, and 370).

Line 106: Typo of the word “enrollment”

Thank you. The typo has been amended (Line 126).

Line 109: What is meant by “Follow-up ended invariably on 22 August 2021”. Why? Please explain. Was this a pre-determined date? 

Thank you for your comment. We meant that all patients were followed at most until 22 August, because it was the date of the last data available. This is the reason why we enrolled patients until 22 July, to have at least one month of follow-up to measure the outcomes. We have rephrased the text (Lines 129-130).

Lines 114-115: You collapsed ethnicity into a dichotomous (native v. migrant) variable. Please include numbers regarding the ethnic groups represented in the study (ns, percentages) in the text of the methods. As migration status may be related to the variables of interest at varying degrees per ethnic group (depending on country of origin, racial tensions, cultural traditions regarding accessing healthcare and/or number of people in household and/or how mental health conditions are viewed, health status mandates for migration, etc.), sensitivity analyses should also be done per ethnic group. 

Thank you very much for your interesting comment. We have included the numbers regarding the ethnic groups in the results section, and in the revised version of the manuscript we consider patient’s nationality as a variable (Lines 216-220). Please, see the table below with all the information and numbers regarding the ethnic groups represented in the study.

Characteristics No mental illness Mental illness Total

Total number of patients, n (%) 103.365 (71.31) 41592 (28.69) 144957

Ethnicity, n (col%) 

Native 82.661 (79.97) 35684 (85.80) 118345 (81.64)

Migrant 20.704 (20.03) 5908 (14.20) 26612 (18.36)

Nationality (n,col%) 

Spain 82.661 (79.97) 35684 (85.80) 118345 (81.64)

Eastern Europe 3746 (3.62) 1349 (3.24) 5095 (3.51)

Asia 767 (0.74) 72 (0.17) 839 (0.58)

North Africa 3000 (2.90) 661 (1.59) 3661 (2.53)

Sub-Saharan Africa 2014 (1.95) 268 (0.64) 2282 (1.57)

Latin America 10408 (10.07) 3254 (7.82) 13662 (9.42)

EU and North America 769 (0.74) 304 (0.73) 1073 (0.74)

As you indicated and it was explained in the discussion, the migratory status may affect the diagnosis of mental illness, which may have an impact on the severity of COVID-19, such as experience of discrimination or access barriers to care (language, culture, legal, etc.). This is why, following your recommendation, it is very interesting to perform sensitivity analyses. We have modified the models of likelihood of mortality and hospitalization adding the patients’ nationality, as you can now see in revised versions of Table 2 and Table 3. We have observed some changes when adding this variable to the models. Ethnicity was significant in the risk of hospitalizations and in the risk of mortality in women, but adding nationality, ethnicity was not significant in any of the four models. Nationality does not contribute with any significant value to the risk of mortality. On the other hand, in the case of hospitalizations, we observed more risk in Latin American men than in Spanish. All the foreign nationalities, with the exception of EU and North America had more risk of hospitalization than Spanish in women. Please see Tables 2 and 3 and Lines 258-260, 268-270, 374-382.

Lines 116-117, superscript d in Table 1: For replication and interpretation purposes, please cite and clearly define the deprivation index utilized in this study. 

Thank you for your comment. This index was specifically developed for Aragon region in the ecological study by Compés et al. using the Population and Housing Census of Aragon. It was calculated at an aggregated level by Basic Healthcare Area according to 26 socio-economic indicators including information on unemployment rates, types of employment, ageing of the population, education, migrants, and housing and neighborhood conditions. We have summarized it in the text (Lines 151-156) and Table 1.

Line 127: Please articulate which diagnostic labels were renamed to facilitate clinical interpretation. 

Thank you for your comment. In this case, “mood disorders” was renamed as “depression and mood disorders”. We have specified it in the text (Lines 168-169).

Lines 146-148: What was your reasoning for conducting all analyses by gender, as opposed to the other variables that you mentioned have previously been indicated as risk factors for poor COVID-19 outcomes (lines 62-65), or the variables with significant p-values in Table 1? If this was pre-determined (and it sounds like it was), please include this rationale in the introduction section. 

Thank you very much for your pertinent comment. It is true that we could have directly performed one single model for the overall population (including women and men) and adjusted it by gender, but we decided to conduct all the analysis by gender (and to include the rest of significant variables in the models as adjustment variables) for two reasons:

- To be in line with the research guidelines of the European Commission which recommends to perform investigations by gender in order to generate gender-specific scientific evidence.

- Because we expected remarkable differences in the prevalence of the different mental health conditions among women and men.

We have introduced the idea of the analysis by gender in the introduction (Lines 82-83) and added the following reference:

- Zachary Steel, Claire Marnane, Changiz Iranpour, Tien Chey, John W Jackson, Vikram Patel, Derrick Silove, The global prevalence of common mental disorders: a systematic review and meta-analysis 1980–2013, International Journal of Epidemiology, Volume 43, Issue 2, April 2014, Pages 476–493, https://doi.org/10.1093/ije/dyu038

Lines 146-147: The phrase, “..examine the likelihood of 30-day all-cause mortality and COVID-19 related hospitalization (dependent variable…” makes it sound like you created some sort of composite of mortality and COVID-19 hospitalization and ran one regression. Please specify that you are describing different models. You should also somewhere in this section (recommended) or elsewhere describe why it is important to look at both COVID-19 hospitalization AND all-cause mortality when drawing conclusions. Please describe how this comparison lends critical information for testing your hypotheses. 

We have rephrased this part to make clear that we ran two models (men and women) per outcome (mortality and hospitalization), four models in total (Lines 194-195).

Regarding the second comment, we have explained more deeply why it is important to study the negative outcomes of COVID-19 in the methodology section: We have analyzed hospitalization and mortality risk in patients with mental health disorders and COVID-19, because these two outcomes define infection severity following the definition of the COVID-19 Treatment Guidelines provided by National Institutes of Health (Lines 130-133). Although it could have been addressed as a single composite outcome (hospitalization and/or mortality), we believed that running the models and presenting the outcomes by separate could add more useful information for the readers depending their outcome of interest.

Line 150: How did you determine which covariates to include in the models? 

We included those sociodemographic covariates relevant for the study and available in our cohort (age, deprivation index, migrant vs native, rural vs urban) and multimorbidity and polypharmacy as clinical covariates in addition to each type of mental illness as a measure of burden of diseases and treatments. Unfortunately, we were not able to include other relevant variables for the study because they were not available in our cohort, as it is explained in the limitations section (Lines 435-437).

Statistical Analysis section: How were missing data handled? 

The information regarding socio-demographic factors was complete for all the participants since the recording of these variables is mandatory in EHRs, so we did not have missing data. In regard to clinical variables, mental health conditions and chronic diseases (for number and definition of multimorbidity) were assumed not to be present in a patient if they were not recorded in their EHRs, and their absence was not considered as missing data in any case.

Table 1: I appreciate the inclusion of the percent representation for the migrant group within the overall category of individuals with schizophrenia. Please include additional clarification on the percent representation specifically within the migrant population. For instance, could you provide the percent of individuals with schizophrenia among migrants, based on the total number of migrants? This clarification would enhance the interpretation of the data and contribute to a more nuanced understanding of the prevalence within the migrant subgroup. I have the same comment for the other characteristics listed in Table 1. 

Thank you for your comment. We have added in Table 1 the information of patients depending of their nationality. We have also clarified in the table that the percentage of native and migrant is per column.

We calculated the prevalence in number and percent of the individuals with each mental health disorder based in the total number of migrants, and we have added this information in the results section (Lines 220-222). Please, see below the table with all the information. 

Total number of migrants 26612

Mental illness, n (col%) 

Schizophrenia spectrum disorders 57 (0.21)

Anxiety disorders 4203 (15.79)

 Cognitive disorders 412 (1.55)

Depression and mood disorders 1886 (7.09)

Substance abuse 369 (1.39)

Personality disorders 62 (0.23)

Eating disorders 90 (0.34)

Lines 176-177: Please also include the ns and percentages for those without mental health diagnoses and COVID-19. 

We observed that 1,386 (1.3%) patients without mental illness died and 7,018 (6.8%) were hospitalized. Lines 230-231.

Line 197: Please double check this sentence. Did you mean that “None of the mental health diagnoses studied affected the likelihood of COVID-19-related hospitalization in men.”? 

Yes, we meant this, since none of the mental health conditions had a significant p value regarding hospitalization outcome. We have rephrased the sentence as suggested (Lines 253-254).

Lines 228-231: I appreciate that you have described some of particulars related to migrant status as related to the “healthy migrant effect”. Please also provide alternate explanations for the migrant findings. 

Thank you for your comment. Another explanation of the lower diagnosis of mental health disorders among the immigrant population is also related to immigrants´ health disadvantage explained by their experience of discrimination, access barriers to care (language culture, legal, etc.) and directly to the inherent stress of being an immigrant (Lines 293-296).

Discussion: For men, living in most deprived areas decreased mortality risk (14.5%) but living in a less deprived area increased hospitalization risk (11%). And being a migrant increased hospitalization risk by 74%. For women, being a migrant decreased mortality risk substantially (nearly 50%), but increased hospitalization risk by 90%. And living in a less deprived area increased hospitalization risk by 12%. These findings and potential implications need to be expounded on in the discussion. 

Thank you very much for the comment. With the change in the likelihood models with the inclusion of nationality variable, the ethnicity is not significative, but we observe some differences in the influence of nationality. These changes are noted in Tables 2 and 3 and Lines 258-260, 268-270, 374-379. The increase in risk of hospitalization in the migrant population classified by nationality has been an unexpected outcome. In general, other studies have shown that this population suffers fewer hospitalizations than the native population. This finding would merit specific studies aimed at finding a more detailed explanation. Please see lines 379-382.

About the influence of deprivation index in the risk of mortality and hospitalization, we observed differences in the models analysed. The risk of hospitalization was higher in patients living in less deprived areas. This could be explained because migrants may have more difficulties to access the health system, they do not receive adequate care in the initial stages and are treated in more serious situations requiring hospitalization. On the contrary, in this study the risk of mortality decreased in most deprived areas in men. This result must be interpreted with caution since deprivation index is an ecological index calculated in an aggregated form, so it would be necessary to examine more deeply the influence of this variable in mortality risk in specific studies. Lines 383-391.

Lines 246-260: Thank you for detailing some of the previous work on schizophrenia and COVID-19 mortality and related hospitalization. I recommend that you include a sentence or two detailing what your study adds to this literature. 

Thank you very much for your comment. This is one of the few studies that analyze the risk of mortality and hospitalization considering the most frequent patterns of mental illness in the population at the same time, and differentiating by sex (Lines 321-325). We also referenced this fact in the strengths section: Many of the published studies focus on the study of the risk of hospitalizations or mortality, or include one or two mental illnesses. Few studies analyse all factors at the same time (Lines 425-427).

Lines 261-292: From this literature, I am left questioning why you did not include variables related to alcohol and tobacco use, as well as caregiver/family support, and hygiene practices in this study. Please seriously consider including these and if it is not feasible, please describe why and how this may limit the interpretation of the results. I see that you have done so for alcohol and tobacco use and BMI (lines 329-334), but please include other limitations related to lack of inclusion of variables related to caregiver/family support, and hygiene practices. 

Thank you very much. We totally agree that these variables would have been very relevant for our study and for the interpretation of the results. Unfortunately, they were not available in our cohort. We have included this fact as a limitation of our study in the corresponding section (Lines 436-437). We have also added more information in lines 340-341.

Lines 293-296: This sentence, “As we saw, in some cases, multimorbidity….more common in patients with SMI” made me question why you did not delineate specific chronic diseases in this study. I appreciate your reliance on other literature; however, if you want to make this claim (as well as the claim in lines 318-320), I would like to see these analyses broken down by chronic disease type. 

Thank you for your comment. We have carefully revised the text to soften these statements, as it is true that we did not include chronic disease types in the analysis and this was not an objective of our study, although we find the reviewer suggestion very interesting to include these analyses for future projects.

Lines 336-339: Great point regarding the COVID-19 pandemic waves. I also wonder about lockdown/quarantine procedures/recommendations in Aragon, Spain. Please include these details somewhere in the text. This will help with reader understanding of how these results may generalize (OR NOT) to other populations. 

We totally agree that the specific context, measures… in this region during the pandemic are very relevant for the interpretation and generalization of results. 

In Aragon, Spain, different measures of geographic confinement and social restrictions were taken throughout this study period depending on the incidence of cases. In the first three months of the pandemic (March - May 2020) testing capacity in our region was limited at the time and the population was in lockdown. One this period finished, diagnostic tests were performed on all patients who were admitted to hospital and on symptomatic patients, and isolation of infected patients was carried out for 14 days. Contacts of positive patients were also actively studied. The use of masks was mandatory in the whole studied period (Lines 134-141). 

Reviewer #2: Thanks very much for the authors for choosing this important topic.

Well done overall.

However, you may flourish the discussion more around the finding related to males versus female differences. What are the possible pathophysiological, psychosocial explanations as well as implications of such finding?

Dear reviewer,

Thank you very much for your review and your words. One of the main results of this study is indeed the differences observed between men and women. As suggested, we have added in the text a possible explanation for these differences (Please, see Lines 310-317, revised version of the manuscript).

---

## [Editor Report · Decision Letter 1]

22 Jan 2024

Mental health and risk of death and hospitalization in COVID-19 patients. Results from a large-scale population-based study in Spain.

PONE-D-23-37788R1

Dear Dr. Gimeno-Miguel,

We’re pleased to inform you that your manuscript has been judged scientifically suitable for publication and will be formally accepted for publication once it meets all outstanding technical requirements.

Kind regards,

Santiago Gascón, PhD

Academic Editor

PLOS ONE

Additional Editor Comments (optional):

We believe that the authors have responded to the comments made by the reviewers and have changed overlaps with their previous publications, as well as clarifications on funding and accessibility of the study data in the text.

For these reasons (and for the results as a whole), we believe that this is an article that deserves to be published in PLOS ONE.
---

## [Editor Report · Acceptance letter]

2 Feb 2024

PONE-D-23-37788R1 

PLOS ONE

Dear Dr. Gimeno-Miguel, 

I'm pleased to inform you that your manuscript has been deemed suitable for publication in PLOS ONE. Congratulations! Your manuscript is now being handed over to our production team.

Kind regards, 

on behalf of

Dr. Santiago Gascón 

Academic Editor

PLOS ONE